# Hydrogen Stable Isotopes Indicate Reverse Migration of Fall Armyworm in North America

**DOI:** 10.3390/insects16050471

**Published:** 2025-04-29

**Authors:** Eduardo S. Calixto, Silvana V. Paula-Moraes

**Affiliations:** Entomology and Nematology Department, West Florida Research and Education Center, University of Florida, Jay, FL 32565, USA; calixtos.edu@gmail.com

**Keywords:** *Spodoptera frugiperda*, migration, insect resistance risk, biogeochemical marker, dispersal, integrated pest management

## Abstract

Fall armyworm (*Spodoptera frugiperda*) is a destructive pest that causes significant crop damage, especially in the U.S., and has spread to various countries around the world. Understanding how these pests migrate is crucial for predicting outbreaks and developing effective management programs. In this study, we estimated the movement of fall armyworm moths by analyzing hydrogen isotopes in 324 samples collected at the edge of continental U.S., which is considered an interbreeding zone for this species. Our results indicate that fall armyworm moths migrate southward from northern U.S. regions, like the Corn Belt, including states such as Nebraska, South Dakota, Minnesota, Kansas and Wisconsin. This discovery provides important insights into the movement of this pest and potential spread of resistance alleles, which can help improve integrated pest management and insect resistance management. The findings are valuable for developing more targeted and timely pest management strategies to protect agriculture and ensure food security.

## 1. Introduction

Fall armyworm (FAW), *Spodoptera frugiperda* (J.E. Smith, 1797) (Lepidoptera: Noctuidae), is one of the most economically significant pests in the United States. This highly polyphagous insect can feed on more than 350 plant species, showing a particular preference for crops in the family Poaceae, including corn, sorghum, rice, and turfgrass [1]. In 2016, FAW emerged as an invasive pest in regions outside its native range, spreading across multiple continents and causing substantial agricultural losses in the eastern hemisphere [2]. Its sporadic and highly unpredictable outbreaks [3], along with its aggressive feeding behavior, make FAW a key pest of major crops worldwide. Understanding the ecology and migratory patterns of FAW is paramount for forecasting seasonal movements and potential outbreaks, which can aid in developing timely and effective management strategies.

FAW is a highly mobile pest capable of dispersing long distances, with reports of 1600 km in 30 h [4]. Studies have shown that FAW moths migrate northward during the crop season, extending into the northern United States and southern Canada [5]. Since FAW does not enter diapause, two main hypotheses have been proposed to explain its population dynamics. The first, known as the “pied piper” migration hypothesis, suggests a northward movement during spring without a return to southern regions [6]. The second hypothesis, known as reverse (southward) migration, posits a southward movement where moths return to overwintering sites in southern Florida and Texas at the end of the crop season [3,5,7,8]. Although studies have shown support for a northward movement from overwintering regions to northern areas [3,5,7,8,9], there is a lack of supporting data for FAW movement from the northern regions of the U.S. and southern Canada to lower latitudes in North America [10,11].

Evidence for reverse migration has been hypothesized for corn earworm, *Helicoverpa zea* (Boddie), another economic noctuid pest in the U.S. [12,13]. Recent work using stable hydrogen isotopic ratios estimated moths migrating southward as far as the Caribbean basin [14]. This reverse migration phenomenon in *H. zea* together with previous studies on FAW migration suggests that a similar pattern might exist in FAW. For instance, the “pied piper” hypothesis proposes that FAW spreads northward each summer but cannot overwinter [7]; migration could not persist without a return movement, as there would be a potential loss of migratory genes in successive generations [15]; and selection could maintain migration if some individuals return southward [16]. These studies support the possibility of FAW reverse migration, which could have significant implications for integrated pest management (IPM) and insect resistance management (IRM) programs due to the extended persistence of resistance alleles to management tools, such as insecticides and transgenic Bt plants, at a continental scale. Here, we tested the hypothesis of potential reverse migration of FAW in North America by using stable hydrogen isotopes.

## 2. Materials and Methods

### 2.1. Data Collection

A long-term year-round moth trapping program using delta traps (Trécé, Inc., Adair, OK, USA) at the West Florida Research and Education Center, Jay, FL, USA (longitude: −87.143891, latitude: 30.773188) has been established since 2017. This trapping location is considered an intermixing zone at the edge of the continental U.S., where FAW moths from populations in South Florida and South Texas converge during migration [8,9]. The collected moths were stored at under −20 °C. A total of 324 samples of FAW moths were selected for this study, corresponding to about 56 moths per year (Appendix A). Out of these, 19 moths were collected in April, 20 in May, 13 in June, 42 in July, 58 in August, 84 in September, 73 in October, and 15 in November.

### 2.2. Stable Hydrogen Isotopes Analysis

Each moth had the right forewing removed, prepared, and submitted to hydrogen isotopic ratio analyses, as described in Paula-Moraes et al. [14]. The method involves removing scales from the wings using a painting brush and cleaning the wing first with Goo and Adhesive Remover Spray Gel (Goo Gone, CC Holdings, Inc., New York, NY, USA) to eliminate any glue from sticky strips. Then, the wing was submerged in 70% ethanol for 24 h to remove any residual oils on its surface (Appendix A). In the lab, this cleaning method produced the same results as the traditional 2:1 chloroform/methanol treatment (Appendix A, [17]). When dried, wings were cut into small pieces (range: 0.084–0.191 mg, Appendix A) and submitted to hydrogen isotope analyzes in the Stable Isotope Mass Spec Lab, University of Florida, Gainesville, FL, USA. Samples and standards were analyzed in a Thermo Electron DeltaV Plus isotope ratio mass spectrometer coupled with a ConFlo IV interface linked to a TCEA (high-temperature conversion elemental analyzer). After weighing and loading the samples and standards into 4 mm × 6 mm silver capsules, samples were left in 96-well plates for 48 h. This process ensures that isotopic composition of the samples and standards remained comparable and consistent [18]. To determine nonexchangeable hydrogen, two keratin standards (Caribou Hoof Standard—CBS, and Kudu Horn Standard—KHS) were used.

After placing the capsules into a zero Blank autosampler at 1400 °C, hydrogen isotopic values (δ^2^H) were measured in a Picarro L2120-I isotopic liquid water and water vapor analyzer (Santa Clara, CA, USA) coupled with a Picarro A0211 high precision vaporizer and a CTC HTS PAL autosampler (Santa Clara, CA, USA). Precision was based on USGS42 = 2.99 ‰ (*N* = 11). To standardize the results, two internal University of Florida water standards (UW Antarctic water and Lake Tulane water) were used, which were calibrated using international standards (USGS49 and USGS50). Isotope results are reported in standard delta notation relative to Vienna Standard Mean Ocean Water. Detailed information of all the process can be found at Paula-Moraes et al. [14].

### 2.3. Inferences on the Probability of Origin

Probability of origin of each sample was inferred based on the methodology described in Ma et al. [19] using the package assignR [19] in the R software version 4.3.1 [20]. First, we built an isoscape based on the hydrogen isotope values of amount-weighted, growing-season precipitation at 5 arc-minute resolution. The isoscape was then calibrated using known values of hydrogen isotopes retrieved from published literature that were included in the assignR package [19]. The package contains a database with hydrogen isotope data from known-origin samples (sample values of *Danaus Plexippus*, monarch butterfly, already available in the R package). Then, calibration was performed to transform all hydrogen isotope data onto a common reference scale, ensuring comparability across datasets from different laboratories. We used the isotope values from the monarch butterfly database, a Lepidopteran species, to provide a realistic comparison of hydrogen isotope assimilation from the environment into wing tissues, ensuring consistency in taxon, geographic region, and analytical approach [21]. Then, a linear model between the environmental (precipitation) isoscape values and the known values (monarch butterfly values) was fitted to produce a calibrated isoscape. With the calibrated isoscape, we generated posterior probability maps for each unknown sample (i.e., field-collected samples) using the Bayesian inversion method. This approach calculates the scaled probability of origin for each grid cell and produces a raster map. Each cell’s value represents the probability that it is the actual origin of the sample among all cells in the map. Finally, we calculated the average distance and direction based on the potential probabilities of origin. Based on the probability of origin maps, low hydrogen isotope ratios, the month of collection, and moth behavior and landscape, we identified moths likely migrating from northern locations.

## 3. Results

Most of the moths collected showed a high probability of originating in Florida, Texas, and the Caribbean region (see example of a sample in Figure 1). Four samples collected between August and October, the crop season in the northern U.S., had a high probability of a northern U.S. origin (Figure 2). The average distance flown by these moths based on the probability-weighted distance ranged from 1312 to 1897 km (Figure 3). The average bearing ranged from 104 to 140 degrees, potentially indicating a movement toward the southeast from areas with the highest probability of origin (Figure 4). This suggests a potential movement from north Texas and Oklahoma to upper Midwest and Corn Belt region, such as North Dakota, South Dakota, Minnesota, Nebraska, Iowa, and Kansas. Other regions, like the northeast and parts of the southern U.S., also showed some probability of origin (Figure 2).

## 4. Discussion

Our results add new evidence for the reverse migration of FAW in North America [5,10,11], and this is the first study to support a large-scale southward movement of FAW using a biogeochemical marker. Based on the hydrogen isotopes ratio analysis, we identified a high probability of origin of moths in the northern region of the U.S., with a potential origin in the Corn Belt, the upper regions of southern states, and the northeast. Molecular studies have supported the northward movement of FAW due to the genetic similarity between populations in the northern and southern U.S. associated with prevailing wind patterns [5,8]. However, limited research has provided compelling evidence for the reverse migration at the end of the crop season [10,11]. Stable hydrogen isotope analysis of FAW wings offers valuable information on the geographical origins of larval feeding, shedding light on the migratory dynamics of this pest. This southward movement was already expected due to the lack of studies supporting the “pied piper” hypothesis, the potential loss of migratory genes present in successive populations migrating northward, and the evidence for reverse migration in other noctuid species in the U.S., particularly *H. zea* [12,13,14].

The probability maps suggest potential moth origins in the Corn Belt, upper regions of the southern states, the northeast, and some parts of the west. The rose diagrams and kernel density functions for distance provide valuable insights into migration direction and distance. However, these results must be interpreted alongside the species’ ecology and landscape characteristics. Few studies have examined FAW migration across the Rocky Mountains (see [8]), where our δ^2^H-based analysis also indicates a high probability of origin. Given the known ecological constraints and landscape barriers, we consider western origins unlikely despite some rose diagrams and probability maps suggesting movement from that direction. Instead, the strong overlap between the high probability of origin and the Corn Belt, combined with the timing of moth collection aligning with the crop season in that region, provides robust support for a reverse (southward) migration pattern.

Out of the 324 moths used in our analyses, 88 were collected during October and November from 2017 to 2023 (Appendix A), with all other moths collected from April to September in the same period (Appendix A). Southward movement potentially occurs at the end of the crop season in the northern states, usually around October–November. The low number of moths selected from the end of the crop season used in our study limits the probability of identifying more moths migrating from the northern region. We expected an increase in the number of moths toward the end of the year as the host crops decline and temperatures drop in the northern region. Since we used a broad range of collection months to test our hypothesis, we could evaluate when moths begin migrating southward. The moth with the highest probability of reverse migration based on both probability maps and collection period was sample 239, which was collected in October. Future studies should continue testing the reverse migration of FAW using samples collected, as far as possible, during the end of the crop season in the southern U.S., while also including some samples from earlier months.

One primary limitation of our study is the uncertainty about the very precise origin of migrating FAW moths. While our isotope analysis provides strong evidence for reverse migration, multidisciplinary approaches integrating molecular analyses and additional stable isotopes could enhance our understanding of migration patterns by providing finer resolution on natal origins and dispersal routes [5,8,9,22]. Additionally, although FAW is a polyphagous pest, it has a strong preference for Poaceae, particularly crops such as corn, sweet corn, sorghum, rice, and turfgrass [1]. Since these host plants belong to the same family, and the average fractionations between host plant and lepidopteran moths are around 3‰ [17], the variability in δ^2^H due to plant fractionation is expected to be relatively low among host species. Finally, agricultural irrigation could introduce additional variation in δ^2^H values compared to precipitation-based isoscapes. Although only about 17% of U.S. corn, the primary host plant of FAW, is irrigated [23], but water sources such as groundwater or reservoirs with high evaporation rates could influence δ^2^H signatures. Nonetheless, these effects are likely to be minimal as groundwater and reservoir water typically mix with precipitation over time, reducing localized deviations [24]. These factors can influence δ^2^H signatures, but the broad latitudinal δ^2^H gradient used in our study mitigates the impact of small-scale variations as any minor isotopic shifts are likely overshadowed by the larger geographic patterns [14].

Understanding the movement of FAW moths is a key tool for IPM programs due to their high mobility, migratory potential, and polyphagous feeding habits. Historically, challenges related to the management of FAW are related to the occurrence of outbreaks, which are usually sporadic and unpredictable [3]. Although monitoring systems have advanced in the detection of moths and provided some insights into potential population dynamics of moth pests, e.g., [25,26,27], it is still important to refine forecasting methods and identify migratory routes to enable the timely adoption of management strategies in IPM programs. In addition, this species has evolved resistance to pesticides. For instance, resistance to Bt toxins [28,29,30] and synthetic insecticides [31,32] has already been documented in FAW populations worldwide. Our results indicate that if resistance selection in northern FAW populations acts as a ‘source’ of resistance alleles, then mathematical models should account for evolution of resistance at a continental scale within IRM programs. Overall, our study provides important insights into the population dynamics of FAW in the U.S.

## Figures and Tables

**Figure 1 insects-16-00471-f001:**
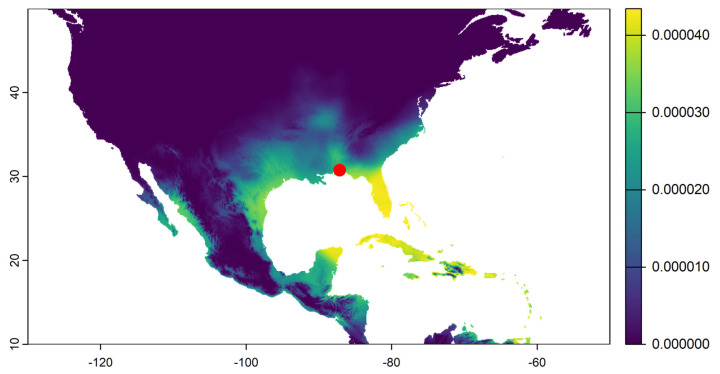
Example probability of origin map of a moth based on hydrogen isotope ratios with high probability of origin in Florida, Texas, and the Caribbean. The yellow color represents a higher probability of origin based on the posterior probability surface of a FAW moth collected at the edge of the continental U.S. See Appendix A. Red dot shows where FAW moths were collected (WFREC, Jay, FL, USA).

**Figure 2 insects-16-00471-f002:**
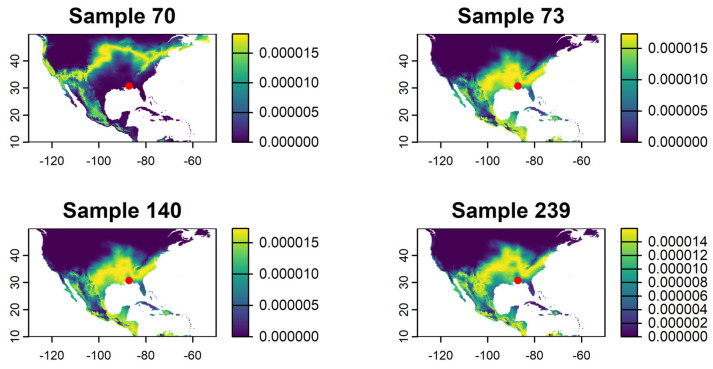
Evidence of FAW reverse migration based on hydrogen isotope ratios (δ^2^H). The yellow color represents higher probability of origin based on the posterior probability surface of a FAW moth collected at the edge of the continental U.S. See Appendix A. Red dots show where FAW moths were collected (WFREC, Jay, FL, USA).

**Figure 3 insects-16-00471-f003:**
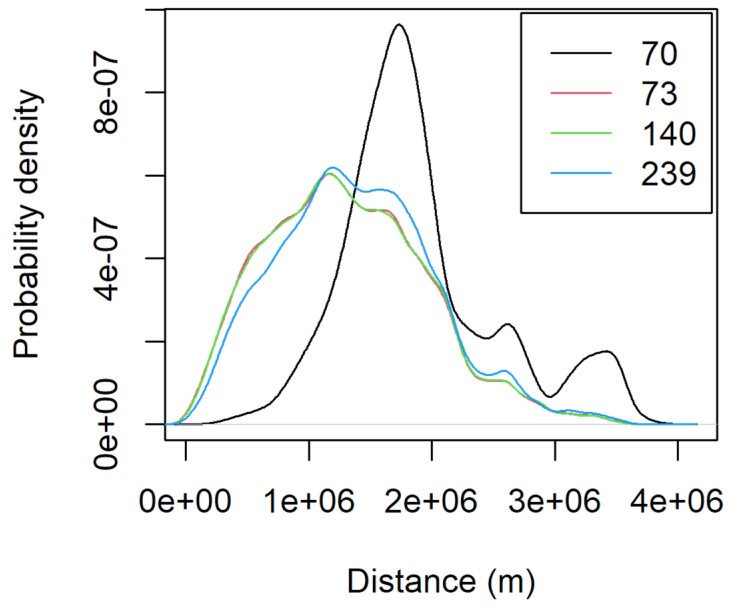
Probability density of the dispersal distance of each moth with a high probability of origin in the northern region of the U.S. based on the posterior probabilities calculated from Figure 2.

**Figure 4 insects-16-00471-f004:**
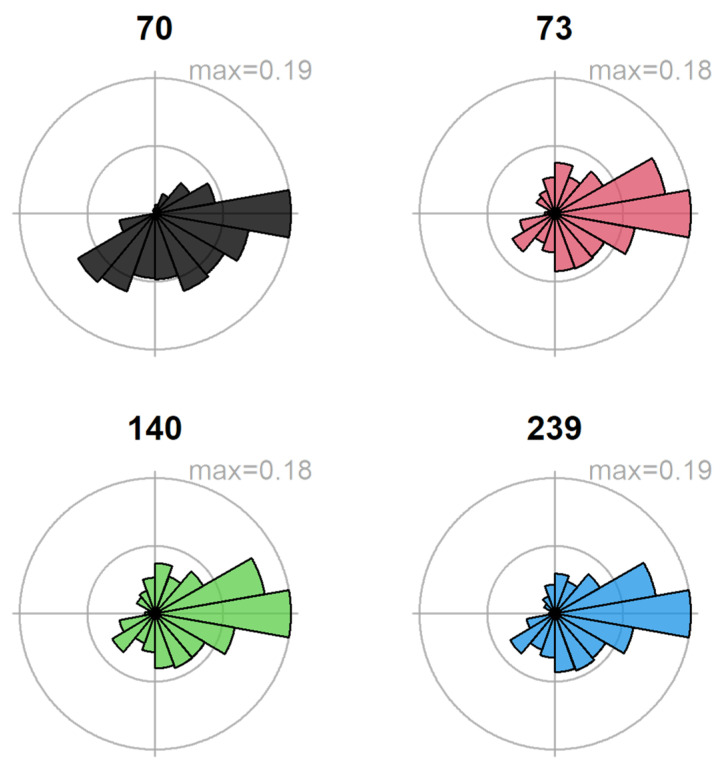
Likely dispersal direction of each moth with a high probability of origin in the northern region of U.S. based on the posterior probabilities calculated from Figure 2.

## Data Availability

The original contributions presented in this study are included in the article/Appendix A. Further inquiries can be directed to the corresponding author.

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
