# Peer review of "Hydrogen Stable Isotopes Indicate Reverse Migration of Fall Armyworm in North America"

_insects, 2025, doi:10.3390/insects16050471_

Round 1

Reviewer 1 Report

Comments and Suggestions for Authors

This manuscript tries to present hydrogen isotopic evidence for southward migration of fall armyworm by analyzing adult samples caught in Florida, US.

However, sufficinet information to support their results is not given in method section 2.3. The section is too simple to understand details (data and methods used) in calibration. I don't know how the authors constructed the probability-of-origin maps with the section's information only, e.g. what are "known values"? What is "published literature"? This is far from the best practice for a scientific paper. Their previous paper (14 in the reference list) presented good information on how they performed calibration. I am surprised at a big difference between the two quality-control levels by the same group. This is my main concering point. I recommend to resubmit a revised manuscript. 

Minor comments.  Figure 1 and 4 are too small to see details. Please make them large and clear with good resolution.

Reviewer 2 Report

Comments and Suggestions for Authors

The authors explore the migratory patterns in the U.S. of a global agricultural pest, the fall armyworm, by using hydrogen isotopes (δ2H) as tracers of insect long-distance movement. Specifically, they test whether captures of fall armyworm close to Pensacola, Florida, show evidence of having grown on the northern region of the North American species distribution and thus have migrated southward (i.e., reverse migration hypothesis). I have outlined below some concerns I have about the methods and the authors’ interpretations, and added some suggestions on how to address them. I think this is an important question and I particularly value the link to IPM and IRM. Valuable inferences can be made from this work, but I believe some further clarification in the manuscript and revisions in the analyses would help either strengthen the argument or at least nuance conclusions.

Major Comments:

-Sample preparation

Removing oils and contaminants from wings is challenging. Particularly if one also must remove glue residues from the sticky strips in delta traps. This can bias the d2H results. Could you clarify:

How does your cleaning procedure compare to traditional 2:1 chloroform:methanol cleaning?

Did you submerge the wings in alcohol, and if so for how long?

Did you do more than one wash, if so, how many and for how long?

Do you have an independent way of confirming this cleaning method was effective?

And could you clarify whether you used methanol or ethanol?

I believe clarifying this should not add more than 1 or two sentences to the manuscript – but the implications of different approaches can be important.

-Analytical methods

The context for the Picarro methods needs to be explained. The sections before gives no indication you will be calibrating to locally collected water sources on top of the precipitation data from assignR  package. Is that the case? If so, please describe and explain the supplementary water δ2H collection. I went to read Paula-Moraes 2024 for clarification. As written now, it would seem to imply you somehow got a reading of the moth d2H through the Picarro.

If you keep this section, consider starting the paragraph with – ‘To measure precipitation (or the source of water used) δ2H values we….’ Many interested readers might not have the background knowledge to know you would only run filtered water through the Picarro.

-Isoscape preparation, assignment information, and interpretation

(1) The section on the continuous surface geographic assignment needs important revisions– without these, it is difficult to evaluate the appropriateness of the methods and data interpretation.

First please add the name of the package in the text – assignR, clarify whether you only used precipitation data from assinR or also included you samples (the point about the Picarro).

How was the precipitation to tissue calibration done? As written here it seems you used all data available in assign R – which would include very different matrices to the one you are interested in, and I would feel reluctant to take this approach (e.g. hair, feathers, etc) – these matrices can behave differently from chitin/wing tissue. I would suggest just using insects here – I think the monarch dataset had a broad spatial coverage (but don’t remember if others are available now).

[Note: I have a similar issue with the fact that the IRMS calibration standards are not the same matrix as your samples, but I am also aware there are no international standards for lepidoptera wings, so I am not suggesting changes on that].

Did you use some of the Moraes et al. 2024 known-origin tissues to calibrate this dataset?  – were any new tissues included here, or did you use the same calibration equation? If so, just mention the calibration equation with the reference and then you don’t need to give as much detail. If you do that, I suggest placing more emphasis on explaining the geographic assignments.  

(2) Based on the caption for Figure 4, I don’t have sufficient information to infer how this probability map was modelled – but I am assuming (given the distribution of values) that likely you used the Joint Probability function - jointP() -in assignR. This type of summary can be desirable, but it has some associated challenges and can lead to misinterpretations. This approach is not appropriate if the samples come from different populations or have different origins (which I think is the case for your fall armyworm) – the reason for this is that it does not weight the values of each independent individual assignment and then multiplies them by each pixel for all individual – easily misrepresenting the joint distribution and probability density.

Potentially, this could be biased further by including in that estimate values from individuals like ID 314, 309, 99, 94, 5 – which have high δ2H values (i.e., -4.1‰, -14‰, -10.5‰, -8.9‰, -16.3‰ respectively). These individuals’ values deviate more from your average δ2H values than your possibly northern origin ones (~ -70‰ to -90‰). I know the interest is on the reverse migration hypothesis – but these high values are also important – and perhaps, better supported as a source of origin.

(3) If a return migration is the focus here – wouldn’t a better test be achieved by collecting only individuals at the end of the production season in the northern regions? I can see the authors acknowledge this in L170-173, but that would have been a better starting point for this research.

I am somewhat surprised that individuals 263, 262 and even 68 could have originated in the northern regions – Duplesis et al. 2020 show that at ~18°C (+/-1°C) the larval and pupation periods take about a month each, so these individuals would have come from eggs laid in ~mid-February. Is this possible? High probability pixels are based on the underlying isocapes and tissue value of the individuals, but they are not indicative of “true” origin, particularly if this would not seem reasonable given other contexts (e.g., temperature).

(4) Since this insect is polyphagous – how do you account for potential important differences in fractionation among different host species or host interaction with those plants? Additionally, is there any additional considerations for divergence between agricultural irrigation water and rain water δ2H? Unlike studies tracing species that ingest wild plants, most likely using rain water for growth, agricultural water can come from multiple sources (e.g., reservoirs with high evaporation or underground water) which could, in turn, deviate significantly from  the predicted δ2H values based on the precipitation isoscapes. Was this taken into consideration? Otherwise, I’d recommend at least addressing it on the discussion.

(5) This point is minor, but I think it goes well with the above comments: would you mind giving a range of values for individuals that originated around the capture site? It would be useful to have a sense of the degree of variation in locally sampled individuals and how far off the presume norther origin migrants are from this range of variation.  

Minor Comments:

-L66 – Could you clarify this sentence? Do you mean moving towards the south? Captures in the Corn Belt at the beginning of the growing season, and coming from the south would instead suggest to me support for the “pied piper” hypothesis.

-L69 – Please cite or clarify what specifically in those previous studies of FAW migration suggest possible ‘reverse migration’.

- L81 – Consider including year range for samples.

-L84 – You are missing an ‘s’

- L93 – Please include the target mass of samples you used.

-L119-120 – How did you determine this? Only by looking at the figures? For example – how confident would you be that individual 262 (one of the northern origin ones with a value of -72.8‰), truly has a more northern origin than individual 8 (not identified as having originated in the north and a value of -69.5‰). This requires a bit more explanation.

-L157 – There’s a typo, I think you meant ‘compelling’.  

-Supplementary Material – I suggest you add mass of sample and % Hydrogen of the sample. Also, change “Hydrogen value” for ‘δ2H (‰)’.

Figures

-Figure 1 – Consider changing the axes units and explaining how to interpret the probability distance curves. I suggest making the maps bigger, they are hard to see clearly. Also please indicate the point of capture in the figure (same for figure 4). See comment on Figure 4 placement.

Figure 3 – I agree that there is indications of movement to the south from a northern origin, but I would argue that movement to the east (from a wester origin) seems to be a stronger inference here.

-Figure 4 – I suggest you consider moving this figure to the first figure position – otherwise it is difficult to visualize the extent to which the individual assignments in current Figure 1 diverge from the other individuals. See major comment on Figure 4 interpretation – I believe this figure needs to be redone.

Round 2

Reviewer 1 Report

Comments and Suggestions for Authors

I thank the authors for their revising efforts. The manuscript has been properly revised according to the reviewers’ comments. The study's logic and results are now clear, the discussion has been much improved. I enjoyed reading the revised version. Before submitting the final manuscript, please correct “analyzes” in line 104, and “double spaces” in e.g. line 268 in discussion section.

I recommend that the manuscript will be accepted.